# Physiotherapists' and occupational therapists' perspectives on information and communication technology in stroke rehabilitation

Mille Nabsen Marwaa[1]*, Hanne Kaae Kristensen[2], Susanne Guidetti[3], Charlotte Ytterberg[2,3,4]

1 Department of Physiotherapy Education, University College Southern Denmark, Esbjerg, Denmark,
2 Research Unit in Rehabilitation Institute of Clinical Research University of Southern Denmark, Odense, Denmark, 3 Department of Neurobiology Care Sciences and Society, Karolinska Institutet, Huddinge, Sweden, 4 Function Area Occupational Therapy and Physiotherapy, Karolinska University Hospital, Stockholm, Sweden

* mnma@ucsyd.dk

**Data Availability Statement:** All relevant data are within the manuscript.

**Funding:** This study has been supported by University College South Denmark.

## Abstract

### Aim

The aim of this study was to explore the current and potential use of information and communication technology (ICT) to enhance coherent person-centred rehabilitation after stroke, from the perspectives of physiotherapists and occupational therapists.

### Method

Five occupational therapists and four physiotherapists from different phases of the Danish stroke rehabilitation process were included and two focus group interviews were carried out. A grounded theory approach was used throughout the study and a constant comparative method was used in the analysis.

### Results

Three subcategories were identified from the analysis of interviews with participants: 1) ICT and apps as meaningful and supportive in the rehabilitation process, 2) ICT as a tool in communication and documentation and 3) Barriers to the integration of ICT and apps in the rehabilitation process. From these categories one core category emerged: The potential of a personalized app solution to facilitate coherent person-centred rehabilitation.

### Conclusion

ICT was perceived as important to integrate in stroke rehabilitation both for assessment, training and to compensate for remaining deficits. The development of a personalized app solution could accommodate stroke survivors' and significant others' need for insight into and overview over the rehabilitation process as well as access to relevant information, which would thereby empower them. Furthermore, a personalized app solution could also

**Competing interests:** The authors have declared that no competing interests exist.

facilitate follow-up after discharge and was perceived to ease the communication and documentation within and between sectors, as well as communication with both stroke survivors and significant others.

## Introduction

Stroke is the third most common cause of death in the western part of the world, and there are about 15.000 new incidences every year in Denmark [1]. Due to improved medical care and an ageing population, the number of people living with the long-term consequences of stroke, including dependence and reduced participation in everyday life is likely to increase [2–5]. Rehabilitation is a team effort, where health professionals within and between sectors need to cooperate in order to facilitate a coherent rehabilitation process [6–8]. Rehabilitation is essential in order for stroke survivors to maintain, develop and reestablish earlier or new functionalities [8]. Furthermore, it is recommended that stroke rehabilitation should be person-centred, i.e. individualized and tailored to respond to the individual's needs and with the recognition of significant others as crucial actors in the rehabilitation process [9,10]. A person-centred approach to the rehabilitation process has also been shown to promote both stroke survivors' and significant others' active involvement in their treatment and care, thus increasing their empowerment and autonomy [9–12] and reducing rehabilitation costs [7]. Furthermore, specialized rehabilitation programs including physiotherapy and occupational therapy have been shown to be effective in rehabilitation of declined functioning [13]. Due to increased numbers of stroke survivors, shorter hospital stays and more care and rehabilitation delivered in the municipalities [4,14–20] new solutions to support both stroke survivors and significant others in the rehabilitation process are needed [21]. To meet the increasing demands and limit the growth of stroke rehabilitation costs, health services in which information and communication technology (ICT) are used alongside conventional therapy offer new opportunities [13].

### Use of ICT in stroke rehabilitation

ICT, such as mobile phones, computers and tablets are becoming more frequently owned and used among the public, including among older people. ICT devices are also being increasingly used to support stroke rehabilitation and studies have shown that stroke survivors, their significant others, and health professionals are interested in integrating ICT into stroke rehabilitation [13,21–24], given that ICT has the potential to accommodate many unmet needs expressed by stroke survivors and significant others. Among the unmet needs, dissatisfaction with the information level during the inpatient stay [22,25], and a lack of adaptation to changing information needs over time [4,25] have been reported. Studies suggest that ICT has the potential to improve the level and timing of information to patients and their significant others [13], and that individualized information can lead to better quality of life in both stroke survivors and significant others [25,26]. Furthermore, ICT has the potential to increase accessibility to rehabilitation in both rural and urban settings [27,28], reduce travel time and costs [22], increase participation in and adherence to therapeutic activities and to support shared decision-making in person-centred rehabilitation, thereby facilitating better rehabilitation outcomes [4,13,29–31]. ICT solutions also have the potential to provide longer follow-up after discharge from acute care, thus extending the rehabilitation period [13,26,32]. Additionally, the integration of ICT solutions is seen in both local and national visions and strategies to achieve equality and optimize the quality of health care provision, as well as to enhance stroke

survivors' independence and participation in everyday life [33,34]. However, even though a recent systematic review concluded that rehabilitation delivered through ICT beyond acute and subacute care is not inferior to face-to-face rehabilitation, there is still a lack of information regarding cost-effectiveness [35].

Although ICT has been increasingly used in health care in recent years, there has been a lack of involvement of all relevant stakeholders in its development [21]. Many ICT solutions therefore do not meet all stakeholders' needs, including the need for customized ICT solutions for both stroke survivors and their significant others [13,21,36]. Furthermore, there is a lack of qualitative literature identifying physiotherapists' (PTs) and occupational therapists' (OTs) perspectives on the use of ICT in stroke rehabilitation [21,37], including their perspectives on how ICT can support the rehabilitation process and be beneficial for both the stroke survivor, significant others, and health professionals involved in stroke rehabilitation. However, a recent qualitative study has shown that health professionals perceive ICT integration in the rehabilitation process to potentially increase sharing and communication between health professionals and patients, thus facilitating transparency during the rehabilitation process [23]. Recommended areas for future research include continuing to identify gaps along the continuum of the rehabilitation process and to develop new person-centred ICT-supported solutions to fill these gaps and support the rehabilitation process [22] simultaneously involving end-users in all phases of the developments to maximize the usability and usefulness of the ICT-solutions [36,38].

### Aim

The aim of this study was to explore the current and potential use of ICT to enhance coherent person-centred rehabilitation after stroke, from the perspectives of PTs and OTs.

### Study context

This study is part of a larger project aiming to develop ICT solutions that promotes coherent person-centred rehabilitation after stroke, to be used within a range of contexts, nationally and internationally. Coherent rehabilitation is understood as rehabilitation that is coordinated between sectors (i.e. hospitals and municipalities) and where health care professionals mutually cooperate [39].

The project is a collaboration between Karolinska Institutet in Sweden, Makerere University in Uganda, University of Southern Denmark and University College South Denmark. The involvement of end users is a key element of the project. Therefore, several qualitative studies have been performed to explore the experiences of using ICT in everyday life and in rehabilitation after stroke -from the perspectives of stroke survivors, significant others and health professionals in different health care contexts [4,16,23,29]. This present study was carried out to explore the perspectives of OTs and PTs within rehabilitation after stroke in a Danish health care context.

## Materials and methods

To capture the experiences and perspectives of PTs and OTs, qualitative semi-structured interviews were carried out, using a constructivist grounded theory (GT) approach [6,11,20]. In a GT approach data generation is considered to be a construction between the participants and the researcher, where data generation and analysis are carried out simultaneously. The inductive approach to data generation and the analysis allow an investigation of what the participants say, and by using additional elaborating questions, what the participants take for granted. Constructivist GT offers flexible guidelines, instead of rules, recipes and demands.

**Table 1. Characteristics of participants.**

| | Name[a] | Age | Profession | Employment area | Experience in neurological rehabilitation |
|---|---|---|---|---|---|
| **Focus-group 1** | Katarina | 42 | PT | Phase 1 | 9 years |
| | Sanne | | OT | Phase 1 | 15 years |
| | Kasper | 30 | PT | Phase 2 | 1 year |
| | Marie | 42 | OT | Phase 2 | 14 years |
| | Louise | 42 | OT | Phase 3 | 14 years |
| **Focus-group 2** | Trine | 40 | PT | Phase 1 | 10 years |
| | Else | 39 | OT | Phase 1 | 9 years |
| | Kamilla | 33 | PT | Phase 2 | 8 years |
| | Christina | 33 | OT | Phase 2 | 8 years |

[a]Altered names

Phase 1: Acute care and rehabilitation in specialized stroke unit

Phase 2: Subacute care and rehabilitation in specialized stroke units

Phase 3: home-based rehabilitation in the municipality [43]

The research process is not linear and the researcher can adopt all or each of these steps, whether or not theory development intentions are involved [40]. Furthermore, memo-writing and constant comparative analysis of data were used throughout the study [40].

## Participants

PTs and OTs participate in and constitute a great part of stroke survivors' rehabilitation process, such as early screening for motor and cognitive deficits, assessment of stroke survivors' need for support, and goal setting [41,42]. Therefore, the inclusion criteria were PTs and OTs working in various phases of the stroke rehabilitation. In Denmark the stroke rehabilitation process is divided into three phases: phase 1) acute care and rehabilitation in specialized stroke unit; phase 2) subacute care and rehabilitation in specialized stroke units, and phase 3) home-based rehabilitation in the municipality [43]. The participants in this study were recruited from two regional hospitals (rehabilitation phases 1 and 2) and from two rural municipalities, i.e. a geographic area located outside town and cities with low population density (rehabilitation phase 3) in the region of South Denmark.

Since variation is essential in constructivist GT [40], a purposeful sampling strategy was initially used, and participants were included with attention to variation regarding area of employment (rehabilitation phases 1,2 and 3) and education (both PTs and OTs). See Table 1 for participants' characteristics.

## Data generation

Two focus group interviews were conducted in Denmark by the first and second authors. The initial sample involved identifying and selecting PTs and OTs with various experiences in stroke rehabilitation. Two gatekeepers (leaders of rehabilitation departments) purposefully identified PTs and OTs with various experiences of stroke rehabilitation to be included in the study. The focus groups were planned so that both PTs and OTs from different phases of the rehabilitation process after stroke (rehabilitation phases 1, 2 and 3) were represented. However, due to illness and workload on the day of both focus group interviews, one OT and one PT from the municipalities (rehabilitation phase 3) were prevented from taking part, thus phase 3 was represented by only one OT (see Table 1).

Focus group interviews were preferred as a method to gain in-depth information about ICT use in the rehabilitation process, from the OTs' and PTs' perspectives. A focus group interview allows for great dynamics between participants, because they interact, share a wide range of experiences and create new understandings [44–46]. Group interaction in both focus groups was facilitated using three exercises using cards with different open questions: 1) "Please describe a typical rehabilitation process where you work" and "Can you identify any challenges in your daily work?", 2) "Why do you integrate ICT in your work and which tasks demand you to use ICT solutions?" and 3) "Do you have any suggestions about how ICT could support your work and the rehabilitation process?".

Each participant reflected upon the questions for a few minutes, before sharing their experiences and ideas with the group. Each exercise was scheduled to last about 30–40 minutes. Furthermore, findings from our previous studies [4,29] were used to elaborate the interview guide, for example: "Some stroke survivors mentioned using ICT in new ways after their stroke; what experiences do you have regarding this?", or "Stroke survivors and their significant others mentioned the information level as challenging during the rehabilitation process; can you describe if you see any potential for ICT in accommodating this challenge?" Additionally, elaborative questions in the second focus group interview were used to uncover variations and similarities regarding current and potential use of ICT in the rehabilitation process. For example: "Some therapists in the preceding interview mentioned that existing app solutions often present some barriers for stroke patients; what is your perception of this?". The first author was the focus group interviewer and moderator, while the second author had the role of being the focus group observer and asking additional questions [45].

The focus group interviews, which were conducted in October and November 2017, were audio-recorded and transcribed verbatim by the first author and a research assistant. The interviews lasted 91–107 minutes and took place in the afternoons in a conference room at the acute stroke unit (rehabilitation phase 1) and at the subacute stroke unit (rehabilitation phase 2). The therapists working in the same hospital knew each other in advance but did not know the other therapists in the focus group.

All participants received written and verbal information about the study and gave their verbal and written consent to participate. All names were altered to ensure the participants' anonymity. Ethical approval was acquired from the Danish Data Protection Agency (18/60280). Due to the nature of the study, approval by the Research Ethics Committee was not required, in accordance with Danish legislation on research ethics.

## Data analysis

The transcribed interviews were analysed by the first author using a GT approach. The initial coding process was performed to start the process of data analysis. In this phase the researcher inductively generated as many codes as possible, comparing the codes and looking for similarities and differences in data. Important words or sentences were labelled using both descriptive and in-vivo codes. When moving on to focused coding a constant comparative method was used, where codes were related to one another, combined, and sorted to start categorizing the many codes. Data were both compared within and between interviews. The analysis process started after the first focus group interview, in line with GT, and helped to indentify new topics and ideas to elaborate the interview guide in the next focus group interview. Memos were written during the whole process, describing initial reflections after each interview which also assisted the constant comparison process and identified areas for development in the next interview, followed by a more analytic description on how the categories emerged and were

linked together [40]. Throughout the entire process, the results were discussed and reviewed by all authors.

## Results

Three subcategories were identified from the analysis of the interviews with the participants: 1) ICT and apps as meaningful and supportive in the rehabilitation process, 2) ICT as a tool in communication and documentation and 3) Barriers to the integration of ICT and apps in the rehabilitation process. From these categories one core category emerged: The potential of a personalized app solution to facilitate coherent person-centred rehabilitation (see Fig 1).

The PTs and OTs stated that ICT should be an integrated part of the rehabilitation process, given that, today, ICT is deeply integrated in most people's everyday lives and in society in general.

They expressed that they were able to integrate ICT solutions in the assessment and training, because stroke survivors almost always had at least a mobile phone close to them, as both an inpatient and outpatient. Furthermore, they described that many stroke survivors had included ICT management as an explicit personal goal in their rehabilitation plan, to manage everyday life and to be able to compensate for stroke-related deficits. Thus, integrating ICT solutions in stroke rehabilitation was thought to facilitate person-centred rehabilitation.

### ICT and apps as meaningful and supportive in the rehabilitation process

PTs and OTs expressed that they integrated ICT in the rehabilitation process, based on the stroke survivors' experiences of ICT use. OTs, in particular, reported that their initial assessment often included ability to use a mobile phone, rather than other everyday life activities, such as preparing a meal, given that a majority of stroke survivors brought their mobile phones with them on admission to hospital. Managing the mobile phone seemed to be perceived as an important skill to regain after stroke:

*"The patient can relate to not being able to handle their mobile phone, rather than for example not being able to cook"*

*(Sanne, OT, phase 1)*

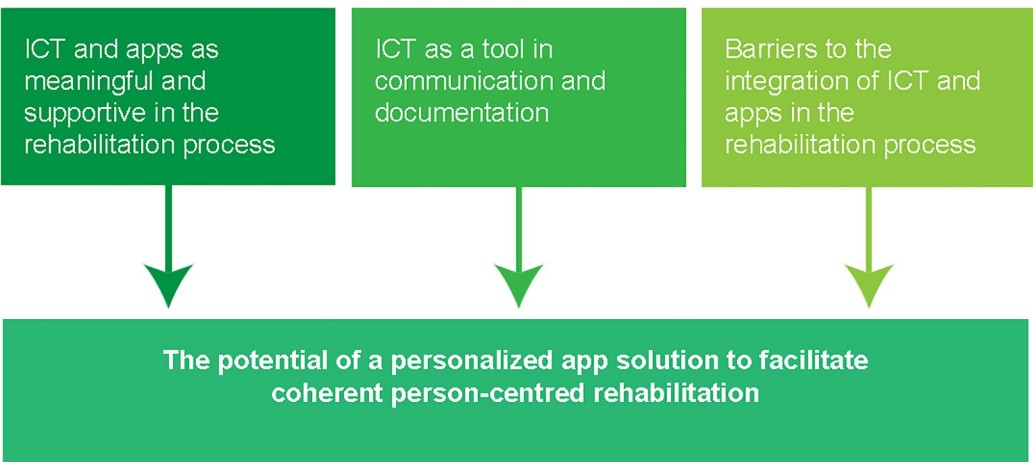

**Fig 1. Overview of the core category and the three subcategories in the results.**

By screening the stroke survivors' ability to use their mobile phone, the OTs got a quick first screening of their cognitive and fine motor skills as she would also have gained using a more time-consuming activity like preparing a meal. The OTs reported being able to screen for memory deficits, for example by observing the stroke survivors' ability to use different passwords (for the mobile phone, e-mail, mobile bank, Facebook, etc.). Furthermore, the ability to send a text message and make a phone call or search the Internet was also often integrated in the early screening process:

*"If the patient has a mobile phone with them on admission we check if they can use it, because then they will be able to call for help when discharged. We also check if they know the different codes on the mobile phone and if they can send a text massage or enter their Facebook account"*

*(Else, OT, phase 1)*

According to the PTs and OTs, stroke survivors who were using their mobile phones in everyday life before the stroke often had ICT management as an explicit goal after stroke. Younger stroke survivors perceived the management of ICT as a precondition for autonomy and freedom, given that ICT use would allow them to do things on their own, as well as retain their role and tasks within the family. The therapists explained that the ability to handle a navigation app on the mobile phone or to call in case of emergency gave the stroke survivor and their significant others a sense of security and at the same time facilitated activities, like taking a stroll:

*"ICT can also be a means to freedom for the patients, for example being allowed to take a walk on their own, where the significant others meanwhile can have a break"*

*(Katarina, PT, phase 1)*

The PTs and OTs explained that they facilitated the stroke survivors to set alarms and reminders, and to dictate messages on the mobile phone, as these app functions could compensate for reduced memory or fine motor skills. Furthermore, for some stroke survivors suffering from fatigue, the pedometer function on the mobile phone was used to visualize the level of activity to achieve a better balance between activity level and energy resources:

*"It [ICT] is a great part of people's lives today and people are depending on it today. By integrating it we can train something useful"*

*(Sanne, OT, phase 1)*

The PTs and OTs reported that they currently integrated a few well-known apps into the rehabilitation process. OTs often integrated apps that could help train facial and speech deficits in collaboration with speech therapists. These apps were reported to be easy to handle for stroke survivors and their significant others. The PTs described that they used an app for recording gait, even if the app had some shortcomings, such as no possibility to place two video recordings next to each other, or to draw lines on the video recordings. These apps could support the stroke survivors to gain an overview of their progress during the rehabilitation process and support them in evaluating the symmetry and quality of their gait. PTs pointed out that it was of great impotance that such a solution was quick and easy if it was to be implemented:

*"I use it to record patients' gait, so they can see their progress, but also to help them see problems with alignment, for example too much weight on one extremity, thereby supporting them visually in trying to correct it. . . it could be great to have a solution where you could show the stroke survivor two recordings next to each other. . . but the solution has to be an easy and quick, or else it will not be used"*

*(Kasper, PT, phase 2)*

The PTs and OTs explained that, when integrating different app solutions in the rehabilitation process, they sometimes had to support the stroke survivor to both install and use the app. Making use of screenshots often supported the stroke survivor to enter and navigate the app. Furthermore, significant others were sometimes involved for support:

*"By taking screen shots you can help the patient remember the different steps in the app. . .I sometimes print these screen shots and give them to the patient, so they can see the next step in the app"*

*(Louise, OT, phase 3)*

When apps were integrated in the rehabilitation process, the OTs and PTs preferred to use an iPad. The use of an iPad was perceived as an excellent way for stroke survivors to regain their cognitive skills; it seemed to be easier than a mobile phone to handle because of the larger screen and was more intuitive and easier to handle than a computer. In addition, the iPad required fewer updates than the mobile phone or computer.

## ICT as a tool in communication and documentation

PTs and OTs reported making use of ICT in different situations during the rehabilitation process, to both document the interventions as well as promote communication with colleagues, the stroke survivor and significant others. However, they also stated that these solutions could be implemented to an even greater extent in the rehabilitation process to accommodate more person-centred rehabilitation.

The PTs and OTs described, for example that they often took pictures with the iPads available on the wards to document how transfers from bed to wheelchair were best performed. These pictures were then printed and hung up by the stroke survivor's bed, to promote communication between health professionals. At discharge, the stroke survivors often took the printed pictures home, thus facilitating continuity in the next phase of rehabilitation. Pictures were perceived as more accurate than solely written guidelines and, furthermore, were time saving for the participants.

Video recordings were also used by some PTs, to communicate with stroke survirvors, significant others and with health professionals in different sectors. They explained that video recordings could promote quality in the rehabilitation process, as they were more detailed and easier to follow than written descriptions and pictures. This created continuity in the rehabilitation process among health professionals both within and between sectors. One PT suggested that these recordings should be done with the stroke survivors' own mobile phone, so they themselves could be the carrier of that information:

*"It would be good if the patient could show the colleagues, this is how you should support me during transfers. If the video recordings are on their own mobile phone or iPad, the patient*

*has the recordings with them when transferring between sectors. It is easier to show instead of only written descriptions from the rehabilitation plan that they receive when discharged"*

*(Trine, PT, phase 1)*

Furthermore, some PTs and OTs described how they used video recordings to document the goals the stroke survivor was currently working on and progress made during the rehabilitations process. Especially when the stroke survivors found it hard to estimate their own progress, video recordings were mentioned to be helpful. This seemed to be valuable for both the stroke survivor and their significant others, as the significant others seldom took part in physio -and occupational therapy, which primarily took place during the daytime. PTs and OTs explained that being able to watch video recordings at the significant others' convenience was a way to stay informed about the stroke survivors' rehabilitation process and it was time-saving for the therapists in comparison to written messages. Additionally, one PT reported that she sometimes used the stroke survivors' mobile phone to record which home exercises she/he had to perform, as the exercises were then tailored to the individual and more accurate, in addition to being time-saving for her:

*"I sometimes video record the patients when doing their exercises, since it is time-saving for me, instead of having to find pictures in our system and print them out. And in this way I record exactly the exercise I want them to do, and don't have to add written instructions. Sometimes I use the patient's own mobile phone for these recordings and sometimes the iPad on the ward. Then the significant others can also see what we have been training, which is very supportive if the patient has aphasia"*

*(Kamilla, PT, phase 2)*

The PT explained that these personalized exercises could then be used by the stroke survivor with support from other health professionals or significant others when training on their own, wheather as an inpatient or outpatient. Another PT pointed out that such a solution could also be a reminder for the stroke survivor to do their exercises, for example she suggested short videos on how to use different devices, and that alarm functionalities could be a support to remember to use the device.

## Barriers to the integration of ICT and apps in the rehabilitation process

Despite PTs' and OTs' willingness to integrate ICT and apps in the rehabilitation process, they described barriers which prevented them from integrating ICT solutions.

One barrier mentioned was that current solutions to attaching video recordings to the rehabilitation plan were time-consuming and complex, and because there were different IT systems in the sectors, attached documents could not be opened in the next rehabilitation phase. Additionally, because of ethical guidelines, the video recordings had to be deleted when the patient was discharged—a step which PTs and OTs indicated they often forgot. Thus, they wished for easier solutions to manage communication within and between sectors:

*"Yes, a way to communicate between sectors with the use of pictures and video recordings, but without us having to move the files from one place to another, but that we can attach them directly into the patients journal in a safe way. That would be worth its weight in gold for many, I think"*

*(Kamilla, PT, phase 2)*

Hence, the PTs and OTs called for solutions where the video recordings could be journalized more automatically, in line with other relevant documentation. They also suggested that video recordings should be easier to access during a busy working day, for example on monitors in the stroke survivors' rooms and/or through personalized app solutions. This would promote the quality of support and communication, as well as save time. The PTs and OTs explained that, when recordings were perceived as time-consuming to access and watch, more conventional solutions like written instructions ended up being more commonly used. This was the case, even though the video recordings and/or pictures as documentation and communication tools were more accurate and easier to follow for the stroke survivor, significant others and health professionals.

A second barrier mentioned by the PTs and OTs was limited access to iPads and computers on the ward, leaving them with reduced options to document interventions right away. They suggested that using an iPad for documentation could be timesaving, as it would enable them to document their findings while performing the assessments. Only one OT from the municipality reported having the opportunity to always bring an iPad when visiting, assessing, and working with the stroke survivor:

> "I can really recommend it, it is definitely time-saving, because then I don't have to write on paper first and later on the computer . . . it´s silly, right? . . .I have also started to search for relevant things together with the stroke survivor right away, for example relevant apps, it is easy and quick"
>
> (Louise, OT, phase 3)

Easier access to iPads could further support the decision about which assistive devices to bring home when the stroke survivor was discharged, since PTs and OTs would be able to show relevant options on the iPad. This could possibly reduce the incidence of inappropriate aids being brought home and be timesaving, in that they did not have to print out different solutions before making a decision:

> "A visual picture gives the stroke survivor a better sense of how the aids looks and if it would be suitable for their home settings; it would make it so much easier if we could go online and find the pictures right away on an iPad"
>
> (Sanne, OT, phase 1)

A third barrier mentioned was the fact that some well-known apps needed a subscription. Some stroke survivors bought the app solution after discharge, but others did not, even if they had benefited from the app earlier in the rehabilitation process. Additionally, many app solutions were not simple enough in their functionalities, and exceeded some stroke survivors' current level of cognitive skills:

> "The patient needs some cognitive resources to be able to manage these things [ICT and apps]. It is a barrier that the patient is only able to see part of the program and must imagine what lies behind the next click. It is part of the initial examination that we find out which abilities the stroke survivor has. Currently there is a lack of programs that are simple enough so that all patients can benefit from apps
>
> (Louise, OT, phase 3)

Furthermore, PTs and OTs mentioned that ICT was regularly updated, sometimes giving the apps a new look and appearance, which confused some stroke survivors:

*"When technologies act unexpectedly the patient is challenged. . . it is important that we have trained the use of ICT many times with the patient, including how to handle these regular updates"*

*(Sanne, OT, phase 1)*

A fourth barrier to integrating ICT solutions in the rehabilitations process, expressed by the PTs and OTs, was the challenge of keeping up with all the different apps available on the market. They often felt prevented from finding alternative solutions to the ones integrated, due to lack of time:

*"We are not good enough to integrate these solutions, since there exists a bunch of apps, and that is the problem, we cannot keep up with it, which apps are relevant and to whom. There are many other things we also have to do in the early phases*

*(Sanne, OT, phase 1)"*

Finally, some PTs and OTs were not confident in using ICT, and because of time constraints in their work, these solutions were currently not being used as much as desired. They called for solutions that were more user-friendly for stroke survivors, such as app templates where they could fill in the details themselves, so that more customized solutions could emerge. Furthermore, they suggested an app solution without any disturbances like advertisements, in addition to being stable and not changing its appearance when updated.

### The potential of a personalized app solution to facilitate coherent person-centred rehabilitation

PTs and OTs described that stroke survivors lacked an overview of their daily activities such as scheduled examinations, therapeutic interventions, times to take medication etc. This lack of overview, especially when as an inpatient, meant having to ask the staff for help to get a general idea of the activities of the day or week. Participants from phases 1 and 2 explained that the interdisciplinary teams had daily morning meetings, where each stroke survivor's activities for the next day were planned and written on a whiteboard in the staff room. They called for solutions where stroke survivors had easier access to their own daily scheduled activities, to avoid the situation where they and their significant others entered the staff room to get a view of the whiteboard, because this violated other stroke survivors' confidentiality:

*"We have these board meetings every day [in the staff room] to get a quick overview of each patient's program of the day. . . but there is a challenge regarding how the patient is gaining access to their own daily program. Currently the patients are not allowed to enter the staff room, so they have to ask the staff or get a quick view from the door. . . there is some development potential there"*

*(Christina, OT, phase 2)*

A solution where all the relevant activities from the whiteboard in the staff room as well as other relevant activities or information could directly reach the individual stroke survivor was

called for by PTs and OTs. They suggested that this could be achieved with a personalized app solution, accessible from the stroke survivors' mobile phone or iPad:

*"I think there is a great potential for an app solution, where health professionals write which activities the stroke survivor has and at what time. . .also what time they have to take their medicine. . . an app that is the stroke survivor's own personalized app in regard to the rehabilitation process, as both an inpatient and outpatient. Many health professionals and significant others calls for such solutions, because stroke patients can have a very complex rehabilitation process"*

*(Sanne, OT, phase 1)*

Furthermore, it was suggested that an app solution contain all relevant information about stroke. In this way all information would be gathered in one place, instead of stroke survivors and significant others having a lot of information on paper or needing to search for information themselves, which would be an advantage for both the stroke survivor and their significant others:

*"Instead of the stroke survivor and their significant other receiving 117 pieces of paper, all information should be gathered in their personalized app solution and you could have a search function to find what you need"*

*(Marie, OT, phase 1)*

At the time of the study, information material about stroke and other relevant information was to be found on the respective hospital's website and on different stroke association websites. However, PTs and OTs experienced that stroke survivors and their significant others often had difficulties searching for and finding relevant information by themselves. They also explained that current information was rarely personalized and not conveniently accessible in a way that would accommodate stakeholders' needs. If relevant information only were to be linked in an app solution and deleted when no longer relevant, it could potentially support the stroke survivor and their significant others in gaining an overview of relevant information during the rehabilitation process. They also suggested that such an app solution should contain answers to frequently asked questions about stroke, from A-Z. Furthermore, they pointed out that such an ICT solution could also prevent documentation from taking place in different IT systems, complicating intersectorial communication and accessibility to important information, such as scan images, rehabilitation plans, transfer pictures and/or videos, etc.:

*"One total solution would benefit both the patients and health professionals. It would be preferable that the health system had fewer places where we have to document. It would be amazing if the systems worked between sectors so that we could communicate more easily with the municipalities and pass on information*

*(Marie, OT, phase 2)*

The PTs and OTs pointed out that such a personalized app solution could offer an overview of the rehabilitation process together with an information and communication platform between the stroke survivors and health professionals, and between sectors. Additionally, the stroke survivor could learn strategies to cope with the complexity of the rehabilitation process, and learn to take responsibility for their activities during the day, instead of taking a more passive and expectant role:

*"I think we sometimes take control of everything when the patient gets admitted, we want to have control over everything, but sometimes the patient involvement gets forgotten, thereby making the patient more passive"*

*(Christina, OT, phase 2)*

To learn strategies, by using ICT to manage their daily lives already during in-hospital rehabilitation, would not only ease the stroke survivors' engagement and empowerment in everyday life in home settings after discharge, but also create a more active, goal-directed collaboration between PTs and OTs, the stroke survivor and their significant others.

## Discussion

PTs and OTs in this study argued that ICT solutions need to be an integrated part of the rehabilitation process, given that ICT nowadays is used in everyday life by most citizens, regardless of age. They perceived ICT integration as important in the early stroke screening, as many urgently need to be able to handle at least their mobile phone again in order to resume previous tasks, feel a sense of security and independence, and to compensate for remaining deficits. These results are in line with previous studies showing that stroke survivors, significant others and health professionals are interested in integrating ICT solutions in the rehabilitation process [4,16,23,28,29,47] because they can support stroke survivors' opportunities to engage in social activities [4,29,48], to give stroke survivors and significant others a greater feeling of independence [4,22,29], and, not least, to improve their quality of life [10,11].

Studies have also shown that health professionals have an important role in promoting successful integration of ICT in the rehabilitation process, as stroke survivors and significant others need to receive relevant introduction to and support on the use of these technologies, including how to handle ICT that acts unexpectedly and for example, needs to be updated [4,13,23,29,49,50]. Thus, therapists in this study pointed out that in order for ICT to be used among stroke survivors it must be easy and simple to handle with attention to cognitive impairments, which is in line with findings from other studies [36,51,52]. A recent study pointed out that stroke survivors with impaired fine hand use may have difficulties managing ICT. However, the majority of stroke survivors and also survivors with aphasia and cognitive impairments can manage ICT with at least one hand [51]. Thus, some stroke survivors having severe cognitive deficits are prevented from making use of ICT solutions, and in these cases more conventional solutions like paper calendars should be used [36,51], which was also expressed by an OT in this study.

PTs and OTs in this study noted that stroke survivors tended to take a passive role during the rehabilitation process, especially at the early phases of rehabilitation. They suggested a personalized app solution to support the stroke survivors and their significant others to gain a greater overview of the rehabilitation process and easier access to relevant information, and to promote collaboration within and between sectors, thereby making rehabilitation more person-centred. Furthermore, OTs and PTs anticipated that with a personalized app solution, the stroke survivor could learn strategies to handle and engage in everyday life after stroke, which would empower them. These findings are supported by another study where the use of ICT is suggested to be effective in empowering people with severe disabilities and their significant others to self-manage their chronic health conditions [27]. Since ICT solutions can be individually tailored, they may contribute to more person-centred rehabilitation [53]. Tailoring information to an individual's needs is important when providing information after a stroke, as needs will differ between individuals and change over time [54]. Other studies also underline

that multidisciplinary team collaboration between sectors may be facilitated by technological solutions [13].

Earlier studies have shown that stroke survivors, significant others and health professionals currently experience insufficient rehabilitation due to lack of time and economic constraints [4,22,27,29]. Rehabilitation today may be too brief to achieve goals, and gains may be lost after discharge due to minimal follow-up [22,29]. Furthermore, both urban and rural populations can be limited in their access to stroke rehabilitation services following discharge due to lack of specialized services nearby, difficulty leaving the home or lack of transportation [22,27,28]. PTs and OTs in this study suggested that the integration of ICT solutions in the rehabilitation process could potentially accommodate some of the stroke survivors' need for continuing access and communication with health professionals after discharge from acute or subacute care. These findings are in line with other studies, that have found ICT solutions to have the potential to create a more extensive follow-up after discharge from acute care, thus extending the rehabilitation period [23,26,32], in addition to being an important safety net and assurance during the initial period of returning home after discharge [28], thus generating feeling of connectedness with the health care system. Furthermore, a more extended follow-up after discharge seems to be motivating for both the stroke survivor and the health professionals [23,49,50,55] and can improve participation in daily life after stroke [56]. Several studies have shown that mobile phone based interventions after discharge, varying from 3–14 phone calls within six months post-stroke, in addition to one personal visit from health professionals, can reduce stroke survivors' and significant others' depression and burden, and improve their health and satisfaction with the rehabilitation process [21,26,28,32,49,57].

Nevertheless, PTs and OTs in this study expressed great challenges in keeping up with the technological landscape, as there are so many rehabilitation apps on the market. They called for simple app solutions which would be free of advertising and aimed at stroke survivors, and that could be personalized more easily than current solutions can. The huge number of existing apps on the market and not being able to individually tailor the app solutions, in addition to many apps requesting payment to acquire full app function, have also been identified as barriers in other studies [13,56,58,59]. Furthermore, many existing app solutions focus on only limited areas of the rehabilitation, e.g., physical or cognitive rehabilitation, and do not embrace the entire rehabilitation process with multiple modalities, including diverse personalized needs. This need for a more comprehensive rehabilitation programme rather than single interventions, was also identified in another recent study [36].

## Strengths and limitations of the study

The inductive approach of this study was suitable, given that the aim was to explore the current and potential use of ICT to enhance a person-centred rehabilitation after stroke—from the perspectives of PTs and OTs [40,46]. The focus group interviews contributed with an in-depth discussion about the possibilities of ICT and the participants reported their thoughts and created ideas and meaning through the dynamics of the focus group interview [46]. To maximize the credibility of the study, participants were selected to represent variation regarding occupation (PT and OT), phases of rehabilitation and experience in stroke rehabilitation. Only two focus group interviews were performed; however the integration of knowledge from our earlier studies regarding stroke survivors' and significant others' use of ICT in the rehabilitation process [4,29] and the use of constructivist GT enabled flexibility in terms of adjusting the interview guide ensured depth in the interviews and a rich data set. Furthermore, to prevent the researcher to ask questions only about known factors, exercise cards with open ended topics and questions were used. However, including health professionals with a greater variation

in age and from different professions (for example nursing staff and medical staff) and a greater representation of therapists from rehabilitations phase three, could have ensured a more thorough and varied description and thereby a deeper understanding, thus achieving a greater credibility and transferability. However, the interpretation of data and the results were discussed continuously between all authors which increased the credibility of this study. In addition, the validity of the results was ascertained by making the steps in the research process transparent to the reader.

## Conclusion

PTs and OTs in this study discussed the importance of integrating ICT solutions in stroke rehabilitation at the early stages of rehabilitation, both to assess stroke survivors' ability to use known ICT but also to train and compensate for remaining deficits. These results—in addition to earlier research within the research group [4,23,29]—have contributed to knowledge about the development of an ICT-based solution to supporting person-centred rehabilitation process for stroke survivors and their significant others. The development of a personalized app solution, that would contain all relevant information needed by the individual stroke survivor and the significant others, thereby giving them a better overview of the rehabilitation process as well as empowering them to take a more active role early in the rehabilitation process, was suggested by the participating OTs and PTs. A personalized app solution could also facilitate a greater follow-up after discharge and was furthermore perceived to ease communication and documentation processes within and between sectors, as well as communication with both the stroke survivor and the significant others. In order to integrate ICT solutions more in the rehabilitation process, organizational barriers must be addressed, e.g., easier access to ICT for PTs and OTs and better IT solutions that allow easy and safe communication between sectors (i.e. privacy protected). The content of a personalized app solution to support a person-centred rehabilitation process is therefore a key area for further research.

## Acknowledgments

The authors thank the PTs and OTs in this study for sharing their experiences and knowledge with us.

## Author Contributions

**Conceptualization:** Mille Nabsen Marwaa, Hanne Kaae Kristensen, Susanne Guidetti, Charlotte Ytterberg.

**Investigation:** Mille Nabsen Marwaa, Hanne Kaae Kristensen.

**Methodology:** Mille Nabsen Marwaa, Hanne Kaae Kristensen, Susanne Guidetti, Charlotte Ytterberg.

**Supervision:** Hanne Kaae Kristensen.

**Validation:** Hanne Kaae Kristensen, Susanne Guidetti, Charlotte Ytterberg.

**Visualization:** Mille Nabsen Marwaa.

**Writing – original draft:** Mille Nabsen Marwaa, Hanne Kaae Kristensen, Susanne Guidetti, Charlotte Ytterberg.

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
