## [Decision Letter · Decision Letter 0]

23 Jun 2020

PONE-D-20-06915

Physiotherapists’ and occupational therapists’ perspectives on information and communication technology in stroke rehabilitation

PLOS ONE

Dear Dr. Marwaa,

Thank you for submitting your manuscript to PLOS ONE. After careful consideration, we feel that it has merit but does not fully meet PLOS ONE’s publication criteria as it currently stands. Therefore, we invite you to submit a revised version of the manuscript that addresses the points raised during the review process.

We look forward to receiving your revised manuscript.

Kind regards,

Imre Cikajlo, Ph.D.

Academic Editor

PLOS ONE

Journal Requirements:

[This study has been supported by University College South Denmark.]

 [The author(s) received no specific funding for this work.]

Additional Editor Comments (if provided):

Please carefully address the issues raised by the reviewers, in particular Reviewer 1.

Reviewers' comments:

Reviewer's Responses to Questions

**Comments to the Author**

1. Is the manuscript technically sound, and do the data support the conclusions?

Reviewer #1: Yes

Reviewer #2: Partly

Reviewer #3: Yes

2. Has the statistical analysis been performed appropriately and rigorously? 

Reviewer #1: Yes

Reviewer #2: Yes

Reviewer #3: Yes

3. Have the authors made all data underlying the findings in their manuscript fully available?

Reviewer #1: No

Reviewer #2: Yes

Reviewer #3: Yes

4. Is the manuscript presented in an intelligible fashion and written in standard English?

Reviewer #1: Yes

Reviewer #2: Yes

Reviewer #3: Yes

5. Review Comments to the Author

Reviewer #1: This is a very well-written article. My biggest concern is that the data are over two years old. In terms of development and speed of advancement of technology, these findings are on the edge of being outdated. This paper is essentially a needs assessment from a larger study and seems to be parsed out from other data. A more robust, comprehensive needs assessment makes more sense rather than just a focus on OT/PT.

Reviewer #2: This is a well written paper with some intersting findings in relation to physitherapist & occupational therapists' perspectives on ICT in stroke rehabilitation. However, there are some fundamental issues that impact on potential to publish. Details as follow:

1. Definition of ICT - the author's correctly note that ICT includes numerous platforms (phone, tablet etc) but then primarily focus on mobile phone use in conjunction with apps. It would be better to specify that the study is focusing on this aspect of ICT and does not address other aspects of ICT such as the delivery of motor rehabilitation via telerehabilitation methods

2. Page 4 paragraph 1 - you mention that "these devices are becoming an integrated part in rehaiblitaton after stroke" - this requires further description and justification.

3. Background - you have not included/discussed Laver et al's Cochrane review on telerehabilitation for stroke - this has some important information/findings on ICT in stroke rehab that you should incorporate

4. Page 5 paragraph 1 - you mention Gustavvson's recent study on the very same topic - how does your study differ? Is this study a replication or do you discuss something different?

5. Page 5 (and throughout the paper) - you use the term "...coherent person-centred rehabilitation" - do you mean cohesive? If you do mean coherent. to whom and what is coherent?

6. Page 6 Participants - description of PTs and OTs as a great part of the rehabilitation process - why select these professional groups and not include nursing, medicine? Are there other allied health professionals who work in the team such as social work or speech pathology? Why were they not included? You must justify the focus on OT/PT particularly given there are some good studies on speech therapist use of ICT for language/communication retraining in stroke rehab

7. Page 8 line 1 - the term 'gaitkeepers' should be 'gate kepers'

8. Page 11 second paragraph - the authors note that initial assessment typically includes the ability to use a mobile phone rather than meal prep - is this more to do with clinician time and/or access to meal prep facilities than the importance of being able to use a mobile phone? Requires clarification/justification

9. Page 13 paragraph 1 - the statement "OTs often integrated apps that could help train facial and speech deficits" - this is quite different to OT roles in other countries / contexts i.e. this type of role is primarily the scope of speech therapists. It would be useful to briefly describe the roles of OT/PT in Denmark if they differ to other countries

10. Page 23 last paragraph "The majority of stroke struvivors... can manage ICT" - this is a very general broad statement made on the basis of a rather small pilot study. Suggest re-wording that there is some preliminary evidence that they may be able to use ICT - please note there was also a review in America that clearly showed many consumers could not easily use apps particularly with mobile phones (ref Boulos, M. N., Brewer, A. C., Karimkhani, C., Buller, D. B., & Dellavalle, R. P. (2014). Mobile medical and health apps: state of the art, concerns, regulatory control and certification. Online journal of public health informatics, 5(3), 229. https://doi.org/10.5210/ojphi.v5i3.4814

11. Page 27 conclusion - please change addressed to discussed. The use of the word addressed suggests the participants were engaged in an intervention rather than discussing experiences, needs, wants.

Reviewer #3: Overall.

This is a well written article. The content provides a comprehensive review of the literature and the current study design.

The results are interesting and relevant to a wide range of rehabilitation specialists. The content provides insight into clinical application and future needs of using ICT for stroke rehabilitation.

Only minor concerns are noted and should be addressed.

Misspellings:

The term wheather throughout should be whether

Gaitkeepers should be gatekeepers

Focusgroup interview page 8 needs space “focus group”

Page 9 “…asking additional questions (43)” needs a period.

Page 19 stabile is stable

Page 24, change an other to another

Overall, typos, formatting, spelling needs to be reviewed again.

A few clarifications of language:

The document uses the term Ipad throughout. This is a specific brand name. It might be better to use tablet, unless, everyone was using an Ipad.

Page 6, please clarify the purpose of the memo-writing.

Page 8, please clarify the process more. There were three exercises using cards with different questions. It is unclear the three exercises. There seems to be three exercises, each lasting 30 minutes, thus an entire session was 91-107 minutes. And 2 of these events occurred for each group or 2 groups. Potentially a diagram of the flow would be helpful.

Page 10, there is more information at this point about the use of memos and the constant comparison. One more sentence to help the readers understand how that works would be helpful.

Page 18, sentence that starts “Easier access to iPads could furthermore…” should be changed from furthermore to further.

Page 19, what does “breaking down” mean, truly not working? Battery life? Not loading?

Page 23, please change “all citizens” to most citizens use ICT.

Page 23, “conventional solutions”, like what?

Page 24. “limited in their access to traditional stroke rehabilitation services”, what are “traditional stroke rehabilitation services”?

Page 25, what are collateral needs?

Page 25, period needed on last sentence.

Page 27, clarify that “safe” means privacy protected (I think).

6. PLOS authors have the option to publish the peer review history of their article (what does this mean?). If published, this will include your full peer review and any attached files.

Reviewer #1: No

Reviewer #2: No

Reviewer #3: No

---

## [Author Response · Author response to Decision Letter 0]

29 Jun 2020

Response to reviewers (point-by-point reply) June 2020

PONE-D-20-06915

Physiotherapists’ and occupational therapists’ perspectives on information and communication technology in stroke rehabilitation

PLOS ONE

Journal Requirements:

Thank you for clarifying this. Font sizes and correct reference style has now been added.

[This study has been supported by University College South Denmark.]

 [The author(s) received no specific funding for this work.]

Thank you for this comment, the section has now been removed from the article. We would like the sentence added to the funding section, thank you.

Point-by-Point reply

Reviewer #1: This is a very well-written article. My biggest concern is that the data are over two years old. In terms of development and speed of advancement of technology, these findings are on the edge of being outdated. This paper is essentially a needs assessment from a larger study and seems to be parsed out from other data. A more robust, comprehensive needs assessment makes more sense rather than just a focus on OT/PT.

Thank you for this comment. Since stroke rehabilitation is a complex area, the decision to only include therapists was to gain insight in defined areas of stroke rehabilitation, which is also mentioned in the introduction as well in the limitation section. 

Furthermore, regarding the data: this study does not aim to provide overview of the ICT available in stroke rehabilitation, but rather how ICT can support stroke rehabilitation in a Danish context, thus empowering stroke survivors as well as their significant others. Therefore, we do not consider data outdated, but still highly relevant how stroke rehabilitation can be supported by integrating different ICT solutions.

Reviewer #2: This is a well written paper with some intersting findings in relation to physitherapist & occupational therapists' perspectives on ICT in stroke rehabilitation. However, there are some fundamental issues that impact on potential to publish. Details as follow:

1. Definition of ICT - the author's correctly note that ICT includes numerous platforms (phone, tablet etc) but then primarily focus on mobile phone use in conjunction with apps. It would be better to specify that the study is focusing on this aspect of ICT and does not address other aspects of ICT such as the delivery of motor rehabilitation via telerehabilitation methods

Thank you for this comment. We wanted to keep the questions broad in the interviews (inductive approach), and during the analysis it was obvious that the mobile phone, iPad and app solutions were most significant and relevant for the therapists to integrate.

2. Page 4 paragraph 1 - you mention that "these devices are becoming an integrated part in rehaiblitaton after stroke" - this requires further description and justification.

Thank you for this comment. The section has now been changed to (page 4):

ICT devices are also being increasingly used to support stroke rehabilitation and studies have shown that stroke survivors, their significant others, and health professionals are interested in integrating ICT into stroke rehabilitation (13,21–24), given that ICT has the potential to accommodate many unmet needs expressed by stroke survivors and significant others.

3. Background - you have not included/discussed Laver et al's Cochrane review on telerehabilitation for stroke - this has some important information/findings on ICT in stroke rehab that you should incorporate

Thank you for this comment. The article has now been included in the background section (page 4)

However, even though a recent systematic review concluded that rehabilitation delivered through ICT beyond acute and subacute care is not inferior to face-to-face rehabilitation, there is still a lack of information regarding cost-effectiveness [35].

4. Page 5 paragraph 1 - you mention Gustavvson's recent study on the very same topic - how does your study differ? Is this study a replication or do you discuss something different?

Thank you for this comment. The recent study of Gusstavsson et al. have some similarities, with conclusion of ICT being able to increase sharing and communication. This study has additional findings in particular the very concrete ideas from the OTs/PTs on how a personalized app solution can embrace the complexity of stroke rehabilitation and empower the stroke survivors and significant others- giving them the tools to manage their own rehabilitation.

5. Page 5 (and throughout the paper) - you use the term "...coherent person-centred rehabilitation" - do you mean cohesive? If you do mean coherent. to whom and what is coherent?

Thankyou for this comment, we have now elaborated this (page 5):

Coherent rehabilitation is understood as rehabilitation that is coordinated between sectors (i.e. hospitals and municipalities) and where health care professionals mutually cooperate [39].

6. Page 6 Participants - description of PTs and OTs as a great part of the rehabilitation process - why select these professional groups and not include nursing, medicine? Are there other allied health professionals who work in the team such as social work or speech pathology? Why were they not included? You must justify the focus on OT/PT particularly given there are some good studies on speech therapist use of ICT for language/communication retraining in stroke rehab

Thank you for this comment. As mentioned on page 5, there is currently a lack of perspectives from OTs/PTs, and as mentioned before, stroke rehabilitation is complex and therefore this step is to gain insight on the ICT use in stroke rehabilitation from a defined area (in this case PTs/OTs) that constitutes a great part of stroke rehabilitation: 

Furthermore, specialized rehabilitation programs including physiotherapy and occupational therapy have been shown to be effective in rehabilitation of declined functioning (13).

Furthermore, there is a lack of qualitative literature identifying physiotherapists’ (PTs) and occupational therapists’ (OTs) perspectives on the use of ICT in stroke rehabilitation (21,37), including their perspectives on how ICT can support the rehabilitation process and be beneficial for both the stroke survivor, significant others, and health professionals involved in stroke rehabilitation

7. Page 8 line 1 - the term 'gaitkeepers' should be 'gate kepers'

Thank you, this has now been corrected: Two gatekeepers

8. Page 11 second paragraph - the authors note that initial assessment typically includes the ability to use a mobile phone rather than meal prep - is this more to do with clinician time and/or access to meal prep facilities than the importance of being able to use a mobile phone? Requires clarification/justification

Thank you for this comment. OTs often use activities of daily living in their early assessments, and this particular OT noticed that some patients see a greater meaning in assessing their ability to handle their mobile phone, since mobile phones are more important for patients than they probably were some years ago. Assessing the ability to use mobile phone gives the OT great insight in cognitive disabilities as would another ADL activity also. We have now elaborated this:

By screening the stroke survivors’ ability to use their mobile phone, the OTs got a quick first screening of their cognitive and fine motor skills as she would also have gained using a more time-consuming activity like preparing a meal.

9. Page 13 paragraph 1 - the statement "OTs often integrated apps that could help train facial and speech deficits" - this is quite different to OT roles in other countries / contexts i.e. this type of role is primarily the scope of speech therapists. It would be useful to briefly describe the roles of OT/PT in Denmark if they differ to other countries

Thank you for this comment. This area is shared/collaboration between OTs and speech therapists. This has now been elaborated: 

OTs often integrated apps that could help train facial and speech deficits in collaboration with speech therapists.

10. Page 23 last paragraph "The majority of stroke survivors... can manage ICT" - this is a very general broad statement made on the basis of a rather small pilot study. Suggest re-wording that there is some preliminary evidence that they may be able to use ICT - please note there was also a review in America that clearly showed many consumers could not easily use apps particularly with mobile phones (ref Boulos, M. N., Brewer, A. C., Karimkhani, C., Buller, D. B., & Dellavalle, R. P. (2014). Mobile medical and health apps: state of the art, concerns, regulatory control and certification. Online journal of public health informatics, 5(3), 229. https://doi.org/10.5210/ojphi.v5i3.4814

Thank you for this comment. This section has now been elaborated and the suggested reference is now included:

…..handle with attention to cognitive impairments, which is in line with findings from other studies (36,51,52).

11. Page 27 conclusion - please change addressed to discussed. The use of the word addressed suggests the participants were engaged in an intervention rather than discussing experiences, needs, wants.

Thank you for this comment, addressed is now replaced with discussed in the conclusion.

Reviewer #3: Overall.

This is a well written article. The content provides a comprehensive review of the literature and the current study design.

The results are interesting and relevant to a wide range of rehabilitation specialists. The content provides insight into clinical application and future needs of using ICT for stroke rehabilitation.

Only minor concerns are noted and should be addressed.

Misspellings:

The term wheather throughout should be whether is now corrected, thank you

Gaitkeepers should be gatekeepers is now corrected, thank you

Focusgroup interview page 8 needs space “focus group” is now corrected, thank you (page 9 and 10)

Page 9 “…asking additional questions (43)” needs a period. is now corrected, thank you (page 9)

Page 19 stabile is stable is now corrected, thank you (page 20)

Page 24, change an other to another is now corrected, thank you (page 24)

Overall, typos, formatting, spelling needs to be reviewed again. Thank you, article is now reviewed for spelling errors again.

A few clarifications of language:

The document uses the term Ipad throughout. This is a specific brand name. It might be better to use tablet, unless, everyone was using an Ipad.

Thank you for this comment. Yes, iPad was specifically mentioned, which is the reason for using iPad instead of tablet.

Page 6, please clarify the purpose of the memo-writing.

Thank you for this comment. The purpose of memo writing is elaborated on page 10:

Memos were written during the whole process, describing initial reflections after each interview which assisted the constant comparison process and identified areas for improvement in the next interview, followed by a more analytic description on how the categories emerged and were linked together (40)

Page 8, please clarify the process more. There were three exercises using cards with different questions. It is unclear the three exercises. There seems to be three exercises, each lasting 30 minutes, thus an entire session was 91-107 minutes. And 2 of these events occurred for each group or 2 groups. Potentially a diagram of the flow would be helpful.

Thank you for this suggestion. We have now elaborated on page 8+9:

Group interaction in both focus groups was facilitated using three exercises using cards with different open questions: 1) “Please describe a typical rehabilitation process where you work” and “Can you identify any challenges in your daily work?”, 2) “Why do you integrate ICT in your work and which tasks demand you to use ICT solutions?” and 3) “Do you have any suggestions about how ICT could support your work and the rehabilitation process?”.

Each participant reflected upon the questions for a few minutes, before sharing their experiences and ideas with the group. Each exercise was scheduled to last about 30-40 minutes.

Page 10, there is more information at this point about the use of memos and the constant comparison. One more sentence to help the readers understand how that works would be helpful.

Thank you for this comment, below is explained that memos worked to both capture initial reflections as well as to use the reflections to constant compare the next interview and between the interviews and to develop the interviews. Minor revisions have been made to this section:

Memos were written during the whole process, describing initial reflections after each interview which also assisted the constant comparison process and identified areas for development in the next interview, followed by a more analytic description on how the categories emerged and were linked together (40)

Page 18, sentence that starts “Easier access to iPads could furthermore…” should be changed from furthermore to further.

Thank you for this comment, this has now been changes to further (page 18)

Page 19, what does “breaking down” mean, truly not working? Battery life? Not loading?

Thank you for this comment, the word breaking down has now been removed, since the meaning lies in being stable and not changing its appearance after updates

…in addition to being stable and not changing its appearance when updated

Page 23, please change “all citizens” to most citizens use ICT.

thank you for this comment, this has now been changed (page 23)

Page 23, “conventional solutions”, like what?

Thank you for this comment, this has now been elaborated: 

….more conventional solutions like paper calendars should be used

Page 24. “limited in their access to traditional stroke rehabilitation services”, what are “traditional stroke rehabilitation services”?

thank you for this comment. We have now clarified (and removed “traditional”) and added “nearby” :

to stroke rehabilitation services following discharge due to lack of specialized services nearby,

Page 25, what are collateral needs?

thank you for this comment. This section has now been changes to (page 26):

….including diverse personalized needs.

Page 25, period needed on last sentence.

Thank you for noticing this. Is now added.

Page 27, clarify that “safe” means privacy protected (I think).

Thank you for this comment. We have now added your suggestion (page 28):

….(i.e. privacy protected).

---

## [Decision Letter · Decision Letter 1]

15 Jul 2020

Physiotherapists’ and occupational therapists’ perspectives on information and communication technology in stroke rehabilitation

PONE-D-20-06915R1

Dear Dr. Marwaa,

We’re pleased to inform you that your manuscript has been judged scientifically suitable for publication and will be formally accepted for publication once it meets all outstanding technical requirements.

Kind regards,

Imre Cikajlo, Ph.D.

Academic Editor

PLOS ONE

Additional Editor Comments (optional):

Reviewers' comments:

Reviewer's Responses to Questions

**Comments to the Author**

1. If the authors have adequately addressed your comments raised in a previous round of review and you feel that this manuscript is now acceptable for publication, you may indicate that here to bypass the “Comments to the Author” section, enter your conflict of interest statement in the “Confidential to Editor” section, and submit your "Accept" recommendation.

Reviewer #1: All comments have been addressed

Reviewer #2: (No Response)

Reviewer #3: (No Response)

2. Is the manuscript technically sound, and do the data support the conclusions?

Reviewer #1: Yes

Reviewer #2: Yes

Reviewer #3: Yes

3. Has the statistical analysis been performed appropriately and rigorously? 

Reviewer #1: Yes

Reviewer #2: Yes

Reviewer #3: Yes

4. Have the authors made all data underlying the findings in their manuscript fully available?

Reviewer #1: Yes

Reviewer #2: Yes

Reviewer #3: Yes

5. Is the manuscript presented in an intelligible fashion and written in standard English?

Reviewer #1: Yes

Reviewer #2: Yes

Reviewer #3: Yes

6. Review Comments to the Author

Reviewer #1: All comments from this reviewer have been addressed by the authors. No further revisions necessary now.

Reviewer #2: (No Response)

Reviewer #3: I have no concerns. Authors addressed all comments sufficiently. The revisions provided improve the work.

7. PLOS authors have the option to publish the peer review history of their article (what does this mean?). If published, this will include your full peer review and any attached files.

Reviewer #1: No

Reviewer #2: **Yes: **Dr Marlena Klaic

Reviewer #3: No

---

## [Editor Report · Acceptance letter]

20 Aug 2020

PONE-D-20-06915R1 

Physiotherapists’ and occupational therapists’ perspectives on information and communication technology in stroke rehabilitation 

Dear Dr. Marwaa:

I'm pleased to inform you that your manuscript has been deemed suitable for publication in PLOS ONE. Congratulations! Your manuscript is now with our production department. 

Kind regards, 

on behalf of

Professor Imre Cikajlo 

Academic Editor

PLOS ONE